# A social network analysis of fraud prediction on crowdsourcing platforms

**Wenjie Zhang**[1,2], **Zhiyuan Nong**[3], **Changyu Hu**[4*]

**1** School of Public Administration, Southwestern University of Finance and Economics, Chengdu, China,
**2** Fintech Innovation Center, Southwestern University of Finance and Economics, Chengdu, Sichuan,
China, **3** School of Management Science and Engineering, Southwestern University of Finance and
Economics, Chengdu, China, **4** College of International Economics and Trade, Ningbo University of
Finance and Economics, Ningbo, China

* huchangyu@nbufe.edu.cn

social network analysis of fraud prediction on
crowdsourcing platforms. PLoS One 21(3):
e0343412. https://doi.org/10.1371/journal.
pone.0343412

Consultancy Services Ltd, UNITED STATES OF
AMERICA

**Peer Review History:** PLOS recognizes the
benefits of transparency in the peer review
process; therefore, we enable the publication
of all of the content of peer review and
author responses alongside final, published
articles. The editorial history of this article is
available here: https://doi.org/10.1371/journal.
pone.0343412

## Abstract

In the context of crowdsourcing contests, where winners take all, attracting high-
quality solvers and solutions presents a significant challenge. A key issue in this envi-
ronment is protecting solvers' intellectual property and preventing fraud risks such as
solution plagiarism and theft. Addressing these challenges is essential for maintaining
the integrity of the platform and encouraging innovation. This study applies social
network analysis to examine the structural characteristics of fraudulent seekers and
investigate whether they exhibit distinct social network features compared to legiti-
mate users. Specifically, we focus on centrality, cohesion, and structural equivalence
to identify potential markers of fraudulent intent. Using a dataset from 9,282 contest
projects initiated in China in 2014, involving 6,241 active users and 246 fraudulent
seekers, we tested a fraud detection model based on social network metrics. The
results reveal significant differences in degree centrality, betweenness centrality,
closeness centrality, and clustering coefficients between fraudulent and non-
fraudulent nodes. The findings demonstrate that social network features, particularly
centrality measures, can effectively differentiate fraudulent seekers from legitimate
users. This study contributes to the theoretical understanding of fraud detection in
crowdsourcing and offers practical insights for the development of more robust fraud
detection strategies.

## Introduction

Crowdsourcing, a key form of open innovation [1–3], enables seekers to post tasks
online to attract independent solvers [4].Crowdsourcing contests, where winners take
all [5], present a significant challenge in attracting high-quality solvers and solutions
[6]. Protecting intellectual property and preventing fraud are critical concerns for
maintaining the integrity of these open innovation platforms [7]. As crowdsourcing
platforms operate in an online, anonymous environment with low entry barriers,

**Data availability statement:** The processed dataset underlying the findings of this study is publicly available in the Figshare repository: https://doi.org/10.6084/m9.figshare.31361668.

**Funding:** This work was supported by the Natural Science Foundation of Sichuan Province, China (Grant No. 24NSFSC3739) and the Ministry of Education Humanities and Social Sciences, China (Grant No. 24YJC630301). The funders played no role in study design, data collection and analysis, decision to publish, or preparation of the manuscript.

**Competing interests:** The authors have declared that no competing interests exist.

safeguarding solvers' intellectual property is increasingly complex [8]. Within this context, seeker fraud emerges as a critical threat. Following platform conventions and prior research [9], we define fraudulent seekers as those who engage in deceptive practices such as "double identity fraud" (creating fake accounts to manipulate contests), "solution embezzlement" (appropriating solver ideas without reward), and "payment refusal" (failing to honor promised compensation). These behaviors breach the core contract of fair exchange. In contrast, non-fraudulent seekers are defined by their adherence to good-faith interaction, including fair evaluation of submissions, respect for intellectual contributions, and timely reward payment. Fraudulent conduct directly undermines solver motivation to invest effort in high-quality solutions, due to the risk of uncompensated work. Meanwhile, the prevalence of fraud discourages legitimate seekers from offering substantial rewards, fearing poor returns. This dual erosion of trust can drive high-skilled talent away from the platform, ultimately degrading the crowdsourcing market into a space for low-level innovation [10].

Despite the challenges, current fraud detection mechanisms tend to focus on protecting seekers' rights rather than those of solvers [11]. While fraud detection techniques, such as fact-checking and relevance assessments [12–14], have been proposed, these methods are often reactive and limited in scope. Furthermore, many existing fraud detection models focus on verbal cues or attribute characteristics, mainly addressing financial fraud [15–17]. However, seekers may not always fabricate false information for monetary gain but may instead seek to obtain solvers' solutions without fair compensation. This highlights the need for more robust detection methods.

To address this gap, we propose using social network analysis (SNA) to detect fraudulent behavior by examining the network structure of users. Social network theory provides a more comprehensive understanding of the relationships and behaviors of individuals within crowdsourcing platforms, making it a promising tool for fraud detection [18]. Scholars have successfully leveraged SNA to detect various fraud types, such as feedback reputation fraud, online auction fraud, and automobile insurance fraud [19–21]. By analyzing how users connect and interact, we can move beyond isolated, attribute-based detection and uncover relational patterns that may serve as stronger indicators of malicious intent.

Consequently, this study employs Social Network Analysis (SNA) to investigate whether fraudulent seekers on crowdsourcing platforms exhibit distinct structural signatures—specifically in centrality, cohesion, and structural equivalence—compared to legitimate users. We hypothesize that these relational differences, captured through a user's position in the social network, can facilitate more accurate fraud detection.

To test this, we model the platform's interaction history as a dynamic social network, where each user is represented as a node and their cooperative or communication relationships form the edges. This network, constructed from 9,282 contest projects in 2014 and encompassing 6,241 users (nodes, including 246 fraudulent seekers), was updated daily to reflect evolving connections. We then calculated and compared key SNA metrics between the fraudulent and non-fraudulent user nodes. The results confirm that fraudulent user nodes occupy significantly different structural



positions. Furthermore, integrating these network metrics into a predictive model substantially improves detection accuracy. This work underscores the value of modeling users as nodes in a dynamic network for fraud detection, offering both practical insights for platform security and a methodological contribution to research in open innovation.

## Research background

### Social network analysis and social networks of online communities

The social network is a comprehensive concept of a collection of internet-based applications in Web 2.0 times [22], which includes social actors and their relationships [23], shaping a network allowing the creation and exchange of User Generated Content [22]. The presence of User Generated Content indicates some interactions of users, such as communication and knowledge sharing. Hence comes the relationship network among them with certain strength and structures. Social network analysis (SNA) is widely used to investigate the characteristics of such relationship network in social activities, including node-leveled discussion and network-leveled discussion [24].

The node-level discussion is mainly conducted focused on centrality analysis [25]. Analyzing the centrality of nodes in the social network, the fraudulent nodes are observed based on the whole network activities, with the network-based characteristics serving as the features of the fraud group to facilitate the fraud detection [26]. In other words, SNA provides a relatively effective way to describe the internal characteristics of social networks and nodes as well as the relationship among them [27].

The SNA explores social relationships hide behind a network, revealing the informational exchange and sharing among nodes [28]. SNA can imitate various kinds of relationships, including terrorist networks and online communities [29]. Classification of nodes based on their network-based characteristics have been widely used in marketing area [30]. By analyzing information or behavior patterns of a large number of nodes, it is possible to develop targeted business marketing strategies [31], such as precision marketing, for specific groups of users. One study on users' behavior shows that core user behaviors could be quickly identified through the process of interaction between users [32]. Another found that the behavioral tendencies of nodes in online social networks are mainly influenced by the similarity of users and link relationships [33].

Scholars have also studied the social networks of crowdsourcing communities. Some explored the impact of different user roles on team performance on the crowdsourcing platform from social networks [34]. Zhang and Wang [35] made a regression analysis by measuring degree centrality, closeness centrality, betweenness centrality, and user contribution to conclude that the network location of users on the Wikipedia platform strongly affects their contribution behavior. Similarly, Lu, Singh [36]' research shows that different nodes in different network locations have different relationship behavior in a popular user support forum. Koch, Hutter [37] used SNA to measure the in-degree and out-degree centrality of nodes in open government platforms as an indicator to classify community user types and found significant differences in the quality of innovation schemes classified by different users.

### Fraud detection method using SNA

The risks brought by the rise of various online Communities urge scholars to do much in-depth research on online fraud detection methods, among which SNA is effective. Ku, Chen [38] combines SNA and decision trees to examine the accuracy of group IDs of a fraudster to detect abnormal relationships in the network, in which network density of subgroups is measured by k-core. Morzy [39] also measured density through SNA and interpret it as a kind of trustworthiness, in which a cluster of participants are found closely connected by committed auctions. Peng, Zhang [40] applied SNA to imitate a relationship network among users in an Internet auction system, providing identification methods to detect fraudsters in real time by filtering transaction data. McGlohon, Bay [41] introduced an algorithm to detect fraud risk in general ledger accounting data based on SNA, the fundamental idea of which is to infer the properties of a user by the properties of other related users. In particular, by observing the behavior of a user's immediate neighbors, it is possible to infer the likelihood

that the user is a fraudster. Yanchun, Wei [42] introduced SNA to analysis fraud behaviors of online feedback, detecting the potential fraudsters in Taobao. Pak and Zhou [43] found that deception leads to a different social structure in computer-mediated communication, suggesting that deception influences most centrality measures and fraudsters show a higher level of cohesion than truth-tellers.

Previous studies have demonstrated that online community users with different network characteristics exhibit different performances and contributions. While SNA has been proven effective in fraud detection [44], most studies focus on criminal gangs or fraudulent accounts rather than specific malicious behaviors. Unlike traditional fraud detection methods that often focus on financial gain, this study highlights that, in the crowdsourcing context, the primary target of fraudulent seekers is the intellectual property of solvers, rather than monetary gain. This study addresses this gap by applying SNA in the context of crowdsourcing platforms, where users form a dynamic social network. The research specifically targets the fraudulent behaviors of seekers, a critical yet underexplored aspect in existing literature.

In the context of crowdsourcing contests, seekers and solvers act as nodes within the network, establishing connections through project initiation and participation. The emergence of fraudulent intentions among seekers is often facilitated by their accumulated experience, and the cost of committing fraud remains relatively low. The innovation of this research lies in its comprehensive examination of the social network characteristics of seekers with fraudulent intentions. By analyzing the structure of fraud nodes as the eigenvalues of fraud groups, we try to uncover unique patterns that differentiate fraudulent seekers from their non-fraudulent counterparts. This novel application of SNA provides a predictive model that can identify seekers with fraudulent intentions in real-time, offering an effective early warning system.

In conclusion, the study significantly contributes to the field by expanding the application of SNA to detect specific malicious behaviors within crowdsourcing platforms. It enhances the understanding of the unique network characteristics of fraudulent seekers, providing valuable insights for developing targeted fraud prevention strategies. This research not only advances the theoretical framework of fraud detection but also has practical implications for improving the security and integrity of crowdsourcing platforms.

## Research hypotheses

Social network analysis has been widely applied in numerous fields to distinguish among categories of network users based on specific characteristics—particularly network structural characteristics—including marketing, information dissemination, influence evaluation, and online fraud detection [45]. These applications rest on the premise that users who occupy different structural positions (represented as nodes in the network topology) exhibit distinct behavioral patterns [46]. Thus, a social network can be understood as comprising two interrelated layers: an abstract topography of nodes (structural positions) and the concrete users who inhabit them. To understand fraudulent seekers on crowdsourcing platforms, it is therefore essential to examine the distinctive nodes they occupy within the network's topography, which differ systematically from those of legitimate users. Centrality, cohesion, and structural equivalence constitute three core network concepts that underpin network effects [47]. Consequently, fraudulent seekers in the social networks of crowdsourcing platforms are likely to possess distinctive network structural characteristics associated with these three core concepts. This study employs social network analysis to examine these characteristic differences and further applies the resultant findings to crowdsourcing fraud prediction.

## Centrality

Centrality refers to the roles of individuals in a network with three specific measurements to indicate structural centrality: degree, betweenness, and closeness [48].

Degree centrality measures the number of direct connections a particular node (user) has in the network [45]. Central users tend to be highly connected, while marginal users have fewer connections [49]. A user with a high degree centrality is considered strategically positioned in the network, facilitating the exchange of information and benefiting from existing

network dynamics [50]. As such, these users are often more trustworthy, as their numerous connections increase the social pressure to maintain an honest image. On the other hand, fraudsters may exhibit lower degree centrality due to their tendency to avoid interactions that could expose their fraudulent behavior.

Betweenness centrality measures the number of shortest paths passing through a node, reflecting the degree of control a user has over information flow in the network [45]. Users with high betweenness centrality occupy critical positions in the network, allowing them to control communication between other users [51]. Users in such positions can manipulate information flow, potentially serving as intermediaries who influence or deceive other users [52]. The higher a user's betweenness centrality, the greater its ability to manipulate information resources, making it an ideal candidate for an intermediary messenger. Fraudsters, who often attempt to manipulate or distort information, may deliberately position themselves in the network to exert control over communication, thereby serving as vital bridges in communication with others.

Closeness centrality measures how efficiently a node can communicate with all other nodes in the network, based on the sum of the geodesic distances between the node and all other nodes [53]. A smaller sum indicates a shorter path, suggesting that the user is less vulnerable to control by other users [54]. This centrality captures two important aspects of a node's role: its control of communication and its efficiency in information flow. The former measures a node's ability to quickly access other nodes, while the latter is positively associated with fewer message transmissions, shorter time, and lower costs in spreading information. Thus, higher closeness centrality implies more efficient access to information and a more central role in the network. Fraudsters, who aim to control the flow of information to obscure their intentions, may seek a more central position, exhibiting higher closeness centrality to manipulate or control communication more efficiently.

Based on the different measurement methods and implications of degree centrality, betweenness centrality, and closeness centrality, each user in a network plays a distinct role in information transmission [55]. Users with higher degree centrality and lower betweenness centrality need to transmit information through some other users [56]; Users in turn with lower degree centrality and higher betweenness centrality may be the key nodes of information transmission in the network; Besides, users with lower degree centrality and higher closeness centrality may be closely related to important nodes in the network, and can also affect and control network resources [43]. Like all strategic behaviors of fraudsters, in crowdsourcing fraud, the seeker with fraudulent intent tries to play a legitimate seeker, while the authenticity of the demand makes it less difficult to conceal and more difficult to detect. Unlike legitimate seekers, who are likely to build broad networks of connections (higher degree centrality) and engage transparently in communication, fraudulent seekers may deliberately avoid forming overtly extensive connections, opting instead for more strategic interactions (lower degree centrality). They are likely to occupy intermediary roles in the network (higher betweenness centrality) to manipulate the flow of information and control the narrative, making it harder to detect fraudulent intentions. Furthermore, fraudulent seekers may exhibit higher closeness centrality, enabling them to maintain efficient access to information and control over key nodes in the network, further complicating detection. Thus, understanding the unique central roles of fraudulent seekers—such as their strategic placement in the network with low degree centrality, high betweenness centrality, and potentially high closeness centrality—may help to significantly improve fraud detection efficiency. Therefore, we propose the following hypothesis:

**Hypothesis 1a (H1a):** There is a significant difference in degree centrality between fraudulent and non-fraudulent user nodes, with fraudulent nodes having lower degree centrality.

**Hypothesis 1b (H1b):** There is a significant difference in betweenness centrality between fraudulent and non-fraudulent user nodes, with fraudulent nodes having higher betweenness centrality.

**Hypothesis 1c (H1c):** There is a significant difference in closeness centrality between fraudulent and non-fraudulent user nodes, with fraudulent nodes having higher closeness centrality.

## Cohesion

The level of interconnections among nodes within a network, known as network cohesion, has been used extensively to identify subgroups or cliques within larger social networks [57]. Research on criminal networks has



revealed that these networks often consist of subgroups of individuals who closely interact with each other. Similarly, in the online auction industry, fraudsters rely on collusive users to improve their reputation scores by creating fake transactions, thereby forming a cohesive group within the social network. Consequently, several studies have concentrated on identifying cohesive groups in social networks [20,58]. Deception's success frequently depends on establishing a "trust" relationship with other members through initial displays [59]. To establish such a relationship, individuals with fraudulent intentions must behave as if they are collaborating with others. However, cohesive subgroup analysis is based on the analysis and metrics of the entire network. In the crowdsourcing social network, unlike other forms of online fraud, the seekers of crowdsourcing are often not group-oriented. As a result, this study focuses on individuals' cohesiveness by utilizing the clustering coefficient rather than cohesive subgroup detection to investigate whether the seekers are in a tighter group. The clustering coefficient describes the probability that neighboring nodes of individuals in the network are also neighbors of one another and is defined as the number of edges connecting the neighborhoods of nodes divided by the total number of possible edges between the neighborhoods of nodes. In crowdsourcing contests, which are typically short-term and discrete tasks, seekers do not require long-term cooperation. However, fraudulent seekers, who tend to have more project experience and a larger network of connections [60], may engage in more strategic interactions, resulting in tighter, more cohesive groups. Non-fraudulent seekers, on the other hand, are less likely to be embedded in such tight-knit groups. Therefore, we propose the following hypothesis:

**Hypothesis 2 (H2)**: There is a significant difference in the cohesiveness (clustering coefficients) between fraudulent and non-fraudulent user nodes, with fraudulent nodes exhibiting higher clustering coefficients.

## Structural equivalence

Structural equivalence is defined as a network property characterizing two or more nodes with highly similar connectivity patterns to all other nodes in the network [61]. It quantifies the positional similarity of network nodes and is widely adopted to identify node sets with prominent within-group homogeneity and clear between-group differentiation.

Structurally equivalent nodes typically connect to overlapping sets of peer nodes, which gives rise to comparable social influence and information exchange patterns; further, such nodes tend to share homogeneous individual attributes (e.g., social status). The higher the structural congruence between two nodes, the greater their tendency to hold consistent initial attitudes toward a specific issue, and the reverse holds true. Scholars investigating network cohesion, structural congruence, and role congruence have documented that nodes occupying the same structural position exhibit consistent behavioral tendencies—a pattern determined by their network location rather than intrinsic individual characteristics [43,62]. Empirical evidence also confirms that users with identical network structural profiles are more prone to behavioral homogeneity than those bound by direct cohesive ties [57,63]. Extending this theoretical logic to the crowdsourcing context, seekers with analogous network structures are thus likely to display similar propensities for developing fraudulent intent. On this basis, we propose the following hypothesis:

**Hypothesis 3 (H3):** The network locations of fraudulent and non-fraudulent nodes exhibit significant differences and lack structural equivalence.

## Methodology

An empirical study was conducted on the crowdsourcing platform to investigate the relationship between centrality, cohesion, and structural equivalence [47]. The study involved four main steps: (1) Constructing relationship networks of user accounts on the crowdsourcing platform based on their cooperation and communication relationships, resulting in a total of 365 networks due to the dynamic daily updates of user networks in 2014. (2) Calculating degree centrality, betweenness centrality, closeness centrality, clustering coefficient, and conducting structural equivalence analysis for each user in each social network. (3) Testing the hypotheses by comparing network metrics between fraudulent and non-fraudulent

nodes. (4) Evaluating the efficacy of social network features in fraud prediction by integrating the network metrics into a random forest-based fraud prediction model.

## Data source and data description

The dataset is from an anonymous, well-known online crowdsourcing platform in China, with approximately 11 million users and releasing more than 460 thousand projects with a total reward of over 400 million dollars. We focus on 9282 contest projects initiated in 2014 with 6,241 active users, including 246 fraudulent seekers. The platform's identification and labeling of fraudulent seekers mainly rely on a post-event, community-driven reporting and investigation mechanism. The platform identifies fraudulent seekers through a three-step, ex-post process: (1) Reporting and Evidence Submission: Solvers initiate the process by formally alleging seeker fraud (e.g., solution embezzlement, payment refusal, or double-identity fraud) and providing supporting evidence; (2) Manual Investigation and Verification: Platform administrators examine the claims, typically by comparing submitted solutions, analyzing user IDs, and reviewing transaction records; (3) Judgment and Announcement: Based on the investigation, a determination of fraud is made and verified fraud cases are publicly announced on the platform's bulletin board. In this study, we leverage the outcome of this process. The publicly listed fraudulent seekers from the platform's official announcements are directly used to label and identify fraudulent accounts in our dataset. Table 1 shows the descriptive statistics of the projects that platform users participated in and the number of messages sent in 2014. On average, each user participated in 1.49 projects and sent 22.42 messages.

Fig 1 shows the number of projects initiated and sent by platform users in 2014. Fig 1(a) shows that the number of projects initiated by a single seeker is about 1–5 in general, the vast majority are below 10, and only a few exceed 10. Fig 1(b) represents that the number of messages these users send to each other is generally less than 20 times, with the minority ranging from 20 times to 40 times and very few exceeding 40 times.

**Table 1. Descriptive Statistics of Projects and Messages in 2014.**

|  | Count | Mean | Std. | Min | Max |
|---|---|---|---|---|---|
| **Projects** | 9282 | 1.490 | 6.350 | 1.000 | 411.0 |
| **Messages** | 139924 | 22.42 | 64.38 | 0.000 | 1882.0 |

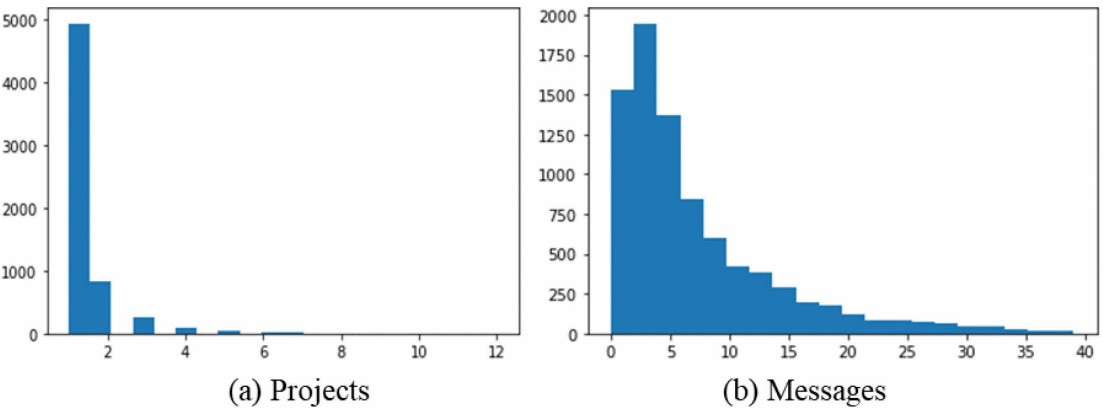

**Fig 1. The number of projects initiated and sent by platform users in 2014.**

## Construct the interaction structure from relationships

The concept of relationship strength is first introduced in the United State, being classified into strong and weak relationships. The latter is believed to be an important information bridge in information transfer, for it is more capable to acquire information and resources over social boundaries than the former [64]. While in China, a typically strong-relationship society versus the US, the strong relationship is believed to be more crucial for acquiring sustainable and powerful help [65]. However, Wellman and Wortley [66] point out that relational connections in social network analysis vary in content and strength, normally interacting in asymmetrical, which means a social network node may be in multiple relationships with various strengths at a time. Taking a relationship between A and B for example. When A and B are a couple of business partners as well as classmates for years, they are in two kinds of relationship – partners and classmates. Moreover, being classmates for years indicates a long-time span of contact, which strengthens the relationship strength between them. Additionally, if A and B get married, they are in a legal relationship, which is a closer relational connection.

In this study, all users on the platform are in two kinds of relationships with different strengths: cooperative relationships and communication relationships. For a cooperative relationship, the user has participated in a project initiated by other users within the last three months and is ultimately selected. This is a strong relationship. For communication relationships, two users have sent messages to each other on the platform. This is a weak relationship. Social ties are inherently dynamic and tend to weaken or dissolve over time without sustained interaction [64]. To accurately capture this evolution, our model dynamically updates the network by incorporating only currently active relationships. The three-month sliding window is adopted based on theoretical, practical, and methodological considerations. Theoretically, it aligns with the sociological understanding that tie strength decays without recent contact. Practically, it corresponds to the typical task lifecycle on the platform, ensuring the network reflects relevant, ongoing collaborations. Methodologically, it balances dynamism and stability—shorter windows may produce sparse, volatile structures, while longer ones risk including obsolete connections, thereby diluting the network's contemporaneity.

Of all 9,282 projects, there are 9,270 projects have completed the bid selection, which means the seeker and the winning solver has reached a cooperative relationship. In other words, there are 9,270 cooperative relationships. Additionally, there is a private message function on the platform. Users communicate through private messages when they feel interested or confused. Since the sending behavior is targeted, a user can send any piece of information to another user to reach a communication relationship. 8858 users sent 208,109 private messages, which means there are 208,109 communication relationships.

A common approach to network construction involves defining edges based on a 'score' that represents tie strength between users [43]. In this study, we treat each user account as a node. A cooperative relationship (score_b) is indicated by a value of 1 if two users have collaborated within the past three months, and 0 otherwise. Similarly, a communication relationship (score_m) is assigned a value of 1 if private messages were exchanged within the same period, and 0 otherwise. To integrate these two types of ties—conceptualized as strong (cooperation) and weak (communication) relationships—we combine their scores to form a weighted edge between nodes. The strong cooperative tie is assigned a weight of 1.0, while the weak communication tie is assigned a weight of 0.5.

The score of the relationship is counted according to formula (1):

$$relationship\ score = score_b + score_m * 0.5 \tag{1}$$

To ensure the robustness of the model and the reliability of its findings, a comprehensive sensitivity analysis will be performed to examine the impact of key parameter choices; specifically, this analysis will assess how variations in the assigned relationship weights (e.g., 0.3, 0.7) influence community detection outcomes, as well as how changes in the temporal window length (e.g., 2, 4 months) affect the resulting network topology and its derived structures.

## Network constructing and analysis

We construct a daily-updated social network model of a crowdsourcing platform using real data to compute node social network metrics and relationships. This model includes seekers and solvers as nodes, and two types of relationships: collaboration and communication. Specifically, the process involves the following steps: First, using the IDs of all seekers of the day as the initial nodes; Second, supplementing new nodes based on the communication and collaboration relationships of the initial nodes over the past three months to form the base network; Finally, connecting all nodes based on their relationships to complete the updated network. The flow chart of network construction is shown in Fig 2.

The network is constructed based on accumulated relationship scores. A line is added when either a cooperative or a communication relationship exists, with the weight increasing discretely in the following order: communication relationship, cooperative relationship, and both relationships. Specifically, the network is updated daily to reflect changes in users' relationships, with a sliding time window of 3 months. As a result, 365 different networks were generated in 2014.

The key metrics used in the model are generally based on the work of Liu et al. [47] and are listed in Table 2. One noteworthy aspect is structural equivalence, which we have improved upon using the work of Pak and Zhou [43]. Since analyzing structural equivalence involves relational analysis rather than just numerical calculation, a significant test would not be meaningful, so we conducted a comparison test instead. We divided the dataset into two groups based on fraudulent projects, resulting in the "fraud & fraud" group and the "fraud & non-fraud" group.

We then performed a Mann-Whitney test on these groups to test for significance in the distance between fraudulent and non-fraudulent nodes. When the p-value is less than 0.05, there is a significant difference in the Euclidean distance between fraudulent and fraudulent nodes, as well as between fraudulent and non-fraudulent nodes. If there is no significant similarity between fraud and non-fraud group nodes but a significant similarity between fraud and fraud group nodes, then hypothesis H3 is supported.

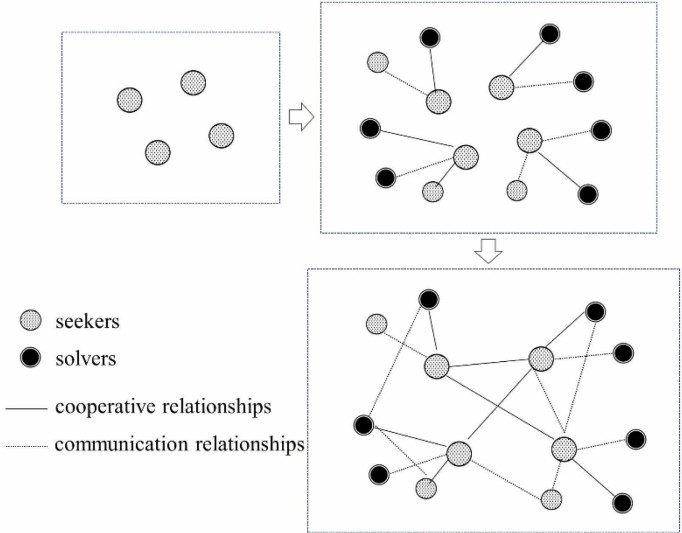

\* The solid lines represent cooperative relationships, while the dotted lines represent communication relationships

**Fig 2. Networking construction process.**

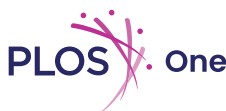

**Table 2. Metrics and equation.**

| Metrics | Equation |
|---|---|
| **Centrality** | |
| Degree centrality | $C_{ADi} = d(i) = \sum_j X_{ij}; C_{RDi} = d(i)/(n-1)$ |
| Closeness Centrality | $C_{ABi} = \sum_{j<k} b_{jk}(i) = \sum_{j<k} g_{jk}(i)/g_{jk}$ |
| Betweenness Centrality | $C_{APi}^{-1} = \sum_j d_{ij}; C_{RPi}^{-1} = C_{APi}^{-1}/(n-1);$ |
| **Cohesion** | |
| Clustering Coefficient | $CC_B = m/C_k^2 = 2m/k(k-1)$ |
| **Structural Equivalence** | |
| Euclidean distances | $dist(X, Y) = \sqrt{\sum_{i=1}^{N} (x_i - y_i)^2}$ |

* The n is the amounts of nodes in the social network; The $b_{jk}$ is the geodesic distance between j and k; The k is the number of all neighboring nodes of node B; The m is the number of edges connecting all neighboring nodes of node B to each other; The N is the dimension of the social network.

## Results

### Descriptive statistics

The dynamic network is updated daily with a sliding 3-month time window over the course of a year, resulting in 365 distinct social networks. For each project, the initiation date is designated as the "Observing Day", on which we crawl the network relationships and calculate the degree centrality, betweenness centrality, closeness centrality, and clustering coefficient for each project seeker. Over the course of the year, there were 138 fraudulent projects and 9,144 non-fraudulent projects. Descriptive statistics are provided in Table 3.

As shown in Table 3, most metrics exhibit significant differences between the fraudulent and non-fraudulent groups. Specifically, the degree centrality, betweenness centrality, closeness centrality, and clustering coefficient of the fraudulent group were lower than those of the non-fraudulent group. We note that the absolute values of these centrality metrics are

**Table 3. Descriptive statistics of centrality and cohesion.**

| Metrics | Count | Mean | Std. | Min | Max |
|---|---|---|---|---|---|
| **Centrality (Fraudulent)** | | | | | |
| Degree centrality | 138 | 0.0000 | 0.0001 | 0.0000 | 0.0009 |
| Betweenness centrality | 138 | 0.0001 | 0.0003 | 0.0000 | 0.0034 |
| Closeness centrality | 138 | 0.0383 | 0.0771 | 0.0000 | 0.2344 |
| **Cohesion (Fraudulent)** | | | | | |
| Clustering coefficient | 138 | 0.0000 | 0.0000 | 0.0000 | 0.0000 |
| **Centrality (non-Fraudulent)** | | | | | |
| Degree centrality | 9144 | 0.0004 | 0.0014 | 0.0000 | 0.0101 |
| Betweenness centrality | 9144 | 0.0011 | 0.0047 | 0.0000 | 0.0380 |
| Closeness centrality | 9144 | 0.0790 | 0.1011 | 0.0000 | 0.2797 |
| **Cohesion (non-Fraudulent)** | | | | | |
| Clustering coefficient | 9144 | 0.0016 | 0.0278 | 0.0000 | 1.0000 |



generally small (e.g., degree centrality ranged only up to 0.0009). This is expected in large-scale, sparse networks like ours, where measures are normalized relative to a large number of possible connections. Interpretation should therefore focus on their relative differences and statistical associations, rather than their absolute magnitude. Notably, the clustering coefficients of the fraudulent group were all 0, indicating that the neighboring nodes of fraudulent ones were not connected, which distinguishes them from the non-fraudulent group. These findings suggest that these metrics exhibit strong discriminant validity and are useful for fraud detection.

### Analysis of centrality and cohesion

Due to the substantial difference in sample size (the non-fraud group is much larger than the fraud group), the T-test is not appropriate. Therefore, the Mann-Whitney U test is used to further assess the significance of differences in the metrics. This test is a widely used and powerful non-parametric method for such situations. The null and alternative hypotheses are based on the assumption that, if the two samples differ, their central locations (i.e., medians) will also differ. The results of this test are presented in Table 4.

As shown, the p-values for degree centrality, betweenness centrality, closeness centrality, and clustering coefficient are all less than 0.001, indicating that these four network characteristics of users with fraudulent intent are significantly different from those of general users at the 0.001 confidence level. To assess the practical magnitude of these differences, we computed Cohen's d effect sizes, interpreting their magnitude using conventional thresholds [67]. For the three centrality measures (degree, betweenness, and closeness), the effects ranged from small to medium (absolute values: 0.21–0.40): degree centrality (d = −0.29) indicates that fraudulent seekers had notably fewer direct connections; betweenness centrality (d = −0.21) reflects their reduced role as information brokers; and closeness centrality (d = −0.40) confirms their greater structural isolation within the network.

For the clustering coefficient, a distinctive pattern emerged: all fraudulent seekers (100%, n = 138) exhibited a value of 0, resulting in zero variance within the group. This complete separation makes standardized mean-difference effect sizes undefined and, more importantly, highlights a fundamental structural difference: fraudulent seekers entirely lacked local triadic closure. In contrast, non-fraudulent seekers displayed a low mean clustering coefficient (M = 0.0016, SD = 0.0278), with values ranging from 0 to 1. The significant U-test and the observed complete separation jointly indicate that the absence of clustering is a robust characteristic of fraudulent accounts in this network. This finding contrasts with existing studies on fraud detection, such as those in the domains of fake reviews and criminal networks [68,69], where fraudulent nodes typically exhibit more interconnected network behavior.

Collectively, these results demonstrate that fraudulent seekers occupy distinct, more peripheral positions in the crowdsourcing platform's social network, with centrality metrics providing meaningful discriminatory power. Thus, Hypothesis 1a was fully supported, whereas Hypotheses 1b and 1c received partial support: fraudulent seekers' betweenness and closeness centrality were significantly lower, not higher, than those of non-fraudulent seekers. In this context, a clustering coefficient of 0 serves as a particularly strong behavioral signature of potential fraud.

**Table 4. Mann-Whitney U test and effect size analysis.**

| Metrics | U-value | p-value | Cohen's d | Magnitude |
|---|---|---|---|---|
| **Centrality** | | | | |
| Degree centrality | 7183.0 | p < 0.001 | −0.2878 [−0.456, −0.120] | Medium |
| Betweenness centrality | 7048.0 | p < 0.001 | −0.2143 [−0.382, −0.046] | Medium |
| Closeness centrality | 7156.0 | p < 0.001 | −0.4038 [−0.572, −0.236] | Medium |
| **Cohesion** | | | | |
| Clustering coefficient | 8418.0 | p < 0.001 | − | − |

## Analysis of structural equivalence

The dataset was divided into two groups based on whether the project was fraudulent or non-fraudulent. The sample size for the "fraud & fraud" group was 9,453 (138*137/2), while the sample size for the "fraud & non-fraud" group was 1,261,872 (138*9,144). Next, the Euclidean distance between every two nodes was calculated based on degree centrality, betweenness centrality, closeness centrality, and clustering coefficient. The descriptive results of the calculated Euclidean distances are presented in Table 5.

To assess whether fraudulent nodes were structurally more similar to each other (structural equivalence), we computed the Euclidean distance between every pair of nodes based on the four network metrics. The pairs were divided into two groups: fraudulent-fraudulent (FF) and fraudulent-non-fraudulent (FN). Descriptive statistics (Table 5) show that the mean distance for FF pairs (M = 0.0627) was smaller than that for FN pairs (M = 0.0900), suggesting that fraudulent nodes may be structurally more similar to one another.

A Mann-Whitney U test indicated a statistically significant difference between the two distance distributions ($U = 4.99 \times 10^9$, $p < .001$). However, because the same node contributes to many pairwise distances, the data lack independence, which can inflate statistical significance. Therefore, we focus our interpretation on effect size rather than statistical significance alone. We quantified the magnitude of the difference using the Common Language Effect Size (CLES) [70]. To align the effect size direction with our research question (i.e., whether FF distances are smaller), we derived the complementary probability, yielding a CLES of 0.582 (Table 5). This value indicates a 58.2% probability that a randomly selected FF pair has a smaller distance than a randomly selected FN pair, corresponding to a small practical effect according to conventional thresholds [67]. In summary, the effect size analysis provides preliminary evidence that fraudulent nodes are slightly more structurally similar to each other than to non-fraudulent nodes. This finding offers directional support for Hypothesis H3.

## Sensitivity analysis

To examine the robustness of our core findings against key methodological choices, we conducted a systematic sensitivity analysis. Two pivotal operational decisions in our study are: (1) assigning communication relationship a weight of half that of cooperative relationship for constructing the weighted network, and (2) employing a 3-month sliding window to build the dynamic network. While these choices are grounded in the platform's project lifecycle and interaction patterns (see Section Data source and data description), it is essential to verify that our findings are not critically dependent on these specific parameter values.

Therefore, we designed a comprehensive test by varying the communication relationship weight across {0.3, 0.5 (baseline), 0.7}and the sliding window length across {2, 3 (baseline), 4} months, resulting in nine distinct parameter combinations. For each combination, we replicated the entire analytical pipeline: network construction, metric calculation (degree, betweenness, and closeness centrality; clustering coefficient), and statistical analysis (Mann-Whitney U tests and structural equivalence analysis).

The comprehensive sensitivity analysis (Tables 6 and 7), which varies the network's edge weight (0.3–0.7) and temporal window (2–4 months), confirms the robustness of our findings. The core descriptive pattern—fraudulent users exhibiting lower centrality and zero clustering—holds universally across all parameter combinations. While the statistical

**Table 5. Descriptive statistics and test results for euclidean distances across groups.**

|  | Count | Mean | Std. | Min | Max | Mann-Whitney U | p-value | CLES(FF<FN) | Magnitude |
|---|---|---|---|---|---|---|---|---|---|
| **Fraud & Fraud group** | 9453 | 0.0627 | 0.0892 | 0.0000 | 0.2344 | $4.99 \times 10^9$ | < 0.001 | 0.5818 | Small |
| **Fraud & Non-fraud group** | 1261872 | 0.0900 | 0.1024 | 0.0000 | 1.0278 |  |  |  |  |

**Table 6. Robustness of network metric difference patterns across parameter combinations.**

| Parameter combination (Weight, Window) | Degree centrality | Betweenness centrality | Closeness centrality | Clustering coefficient | Structural equivalence |
|---|---|---|---|---|---|
| **(0.3, 2 months)** | Fraud<Non-Fraud | Fraud<Non-Fraud | Fraud<Non-Fraud | Fraud=0 | Intra-distance<Inter-distance |
| **(0.3, 3 months)** | Fraud<Non-Fraud | Fraud<Non-Fraud | Fraud<Non-Fraud | Fraud=0 | Intra-distance<Inter-distance |
| **(0.3, 4 months)** | Fraud<Non-Fraud | Fraud<Non-Fraud | Fraud<Non-Fraud | Fraud=0 | Intra-distance<Inter-distance |
| **(0.5, 2 months)** | Fraud<Non-Fraud | Fraud<Non-Fraud | Fraud<Non-Fraud | Fraud=0 | Intra-distance<Inter-distance |
| **(0.5, 3 months)-Baseline** | Fraud<Non-Fraud | Fraud<Non-Fraud | Fraud<Non-Fraud | Fraud=0 | Intra-distance<Inter-distance |
| **(0.5, 4 months)** | Fraud<Non-Fraud | Fraud<Non-Fraud | Fraud<Non-Fraud | Fraud=0 | Intra-distance<Inter-distance |
| **(0.7, 2 months)** | Fraud<Non-Fraud | Fraud<Non-Fraud | Fraud<Non-Fraud | Fraud=0 | Intra-distance<Inter-distance |
| **(0.7, 3 months)** | Fraud<Non-Fraud | Fraud<Non-Fraud | Fraud<Non-Fraud | Fraud=0 | Intra-distance<Inter-distance |
| **(0.7, 4 months)** | Fraud<Non-Fraud | Fraud<Non-Fraud | Fraud<Non-Fraud | Fraud=0 | Intra-distance<Inter-distance |

Note: "Fraud<Non-Fraud" indicates that the mean of the metric for the fraudulent group is consistently lower than that for the non-fraudulent group—a pattern observed in 100% of the combinations. "Fraud = 0" indicates that the clustering coefficient for fraudulent nodes remains zero across all combinations.

**Table 7. Robustness of statistical significance across parameter combinations.**

| Parameter combination (Weight, Window) | Degree centrality | Betweenness centrality | Closeness centrality | Clustering coefficient | Structural equivalence |
|---|---|---|---|---|---|
| **(0.3, 2 months)** | <0.001 | <0.01 | <0.001 | <0.001 | <0.001 |
| **(0.3, 3 months)** | <0.001 | <0.01 | <0.01 | <0.001 | <0.001 |
| **(0.3, 4 months)** | <0.01 | <0.05 | <0.01 | <0.001 | <0.05 |
| **(0.5, 2 months)** | <0.001 | <0.001 | <0.001 | <0.001 | <0.001 |
| **(0.5, 3 months)-Baseline** | <0.001 | <0.001 | <0.001 | <0.001 | <0.001 |
| **(0.5, 4 months)** | <0.001 | <0.01 | <0.01 | <0.001 | <0.01 |
| **(0.7, 2 months)** | <0.001 | <0.001 | <0.001 | <0.001 | <0.001 |
| **(0.7, 3 months)** | <0.001 | <0.001 | <0.01 | <0.001 | <0.01 |
| **(0.7, 4 months)** | <0.01 | <0.05 | <0.001 | <0.001 | <0.01 |

significance level remains at $p < 0.001$ or $p < 0.01$ for the vast majority of tests and shows only minimal relaxation under the most extreme parameter settings, the direction and substantive interpretation of all results are entirely unchanged. This demonstrates that our key conclusions are not driven by specific methodological choices but are reliable properties of the data.

## Experimental evaluation

Prior research [71] investigated the predictive role of various features and their temporal variations in fraudulent intentions across different stages of a crowdsourcing contest; the random forest model employed in this work achieved high accuracy of 79%, 88%, and 98% at the input, process, and output stages respectively, yet exhibited room for improvement in fraudulent intention prediction during the first two stages. To address this, the current study integrates social network features, such as degree centrality, betweenness centrality, closeness centrality, and clustering coefficient, into the fraud prediction model. The goal is twofold: first, to assess the effectiveness of these network features in detecting fraud in practical applications, and second, to enhance the model's predictive accuracy and efficiency by incorporating new features. Through this approach, the study seeks to improve risk control strategies, making the fraud detection model more robust, particularly in the early stages of the contest.

To address the issue of class imbalance and avoid the model's bias towards the majority class, a new balanced dataset is formed by randomly selecting 138 legitimate (non-fraudulent) projects from the original unbalanced dataset. This balanced dataset is then used to conduct robustness tests for the theoretical hypotheses. Subsequently, a model prediction performance improvement validation is performed based on this balanced dataset. The network for fraud detection is constructed using the relationships and interactions among users involved in these projects. The dynamic social network was constructed following the method outlined in Section Network Constructing and Analysis. In brief, each user was represented as a node, and edges were formed based on historical cooperation (score_b) and communication (score_m) relationships within a three-month lookback period. For the subset of users (seekers and winning solvers) involved in the 276 sampled projects, their historical connections were extracted to form the analysis network. Centrality and cohesion metrics were then calculated for each node within this network to characterize its structural position.

After constructing the balanced dataset, the sample is divided into two groups: fraudulent and non-fraudulent. Descriptive statistics and tests for group differences for these groups are provided in Table 8. As shown in Table 8, the mean values for all four network metrics were notably lower in the fraud group compared to the non-fraud group. These findings are consistent with the analysis from the previously used unbalanced dataset, providing initial validation for the robustness of the results.

To assess the statistical significance of the observed differences, Levene's test for homogeneity of variances was first conducted. The results indicated unequal variances for the three centrality measures ($p < 0.05$), warranting the use of Welch's t-test [72]. For the clustering coefficient, which was nearly zero in the fraud group, the Mann-Whitney U test was employed. The test results show that all four metrics exhibit statistically significant differences between the two groups at the $p < 0.001$ level. This provides strong evidence that users with fraudulent intent are fundamentally distinct from other users in terms of network structural characteristics. The consistency of results across datasets (balanced and unbalanced samples) robustly supports the generalizability of these findings.

To enhance the identification of project seekers with fraudulent intent, this study applies social network theory by incorporating four key metrics of social network centrality and cohesion. Two baseline models (BM1 and BM2) were developed using a random forest algorithm and included project-based attributes, seeker-based characteristics, and linguistic cues [71]. As shown in Table 9, BM1 derives its features from the project characteristics at the time of project initiation and the seeker's demographic information (e.g., project budget, project duration, number of friends, and hits). BM2 covers the period up to the day before the winner is announced and focuses on dynamic behaviors of users involved in the project, such as project updates and interactions between the seeker and participants. In particular, various linguistic cues were calculated according to the categories defined by Siering, Koch [15], with fourteen independent variables grouped into six constructs: affect, complexity, diversity, non-immediacy, quantity, and specificity.

**Table 8. Descriptive statistics and tests for group differences.**

| Metrics | Group | Count | Mean | Std. | Min | Max | Levene's test (p-value) | Group comparison (Test statistic, p-value) |
|---|---|---|---|---|---|---|---|---|
| **Degree centrality** | Fraud | 138 | 0.0000 | 0.0001 | 0.0000 | 0.0009 | <0.001 | t=−3.89, p<0.001 |
| | Non-fraud | 138 | 0.0007 | 0.0020 | 0.0000 | 0.0101 | | |
| **Betweenness centrality** | Fraud | 138 | 0.0001 | 0.0003 | 0.0000 | 0.0034 | <0.001 | t=−3.41, p<0.001 |
| | Non-fraud | 138 | 0.0020 | 0.0069 | 0.0000 | 0.0381 | | |
| **Closeness centrality** | Fraud | 138 | 0.0383 | 0.0771 | 0.0000 | 0.2344 | <0.05 | t=−4.44, p<0.001 |
| | Non-fraud | 138 | 0.0863 | 0.1048 | 0.0000 | 0.2575 | | |
| **Clustering coefficient** | Fraud | 138 | 0.0000 | 0.0000 | 0.0000 | 0.0000 | – | U=8418, p<0.001 |
| | Non-fraud | 138 | 0.0007 | 0.0057 | 0.0000 | 0.0667 | | |



**Table 9. Descriptions of features.**

| | | Feature | Explanation |
|---|---|---|---|
| **1st stage (BM1)** | project | pro_total_budget | Reward amount for the project |
| | | pro_duration | Project duration |
| | sponsor | sponsor_truename | Whether the seeker has a real name in the user profile: 1 = Yes, 0 = No |
| | | seeker_gender | gender: 1 = male, 2 = female |
| | | seeker_has_introduction | Whether the seeker has an introduction in the user profile: 1 = Yes, 0 = No |
| | | seeker_hits | Number of clicks that have been won by the seeker |
| | | seeker_group | Whether the seeker participates in the group: 1 = Yes, 0 = No |
| | | seeker_projects_quantity | Number of projects in which the seeker has participated (before the project's release time) |
| | | friends_seeker_quantity | Number of message contacts |
| | Project linguistic cues | linguistic cues* | Same as Mail linguistic cues |
| **2nd Stage (BM2)** | Project | pro_extraintro | Does the project have additional instructions: 1 = yes, 0 = no |
| | | pro_extraintro_length | Supplementary text length |
| | | msg_quantity | Number of e-mail exchanges |
| | | msg_reply | Responder reply time |
| | Mail linguistic cues | Affect_ratio | Ratio of positive and negative words to the total number of words |
| | | Pos_affect | Ratio of positive words to total number of words |
| | | Neg_affect | Ratio of negative words to total number of words |
| | | avg_sentenceLength | Average number of words per sentence |
| | | avg_wordLength | Average number of letters per word |
| | | Pausality | Ratio of the number of punctuation marks (periods, commas, semicolons, colons, question marks, exclamation marks) to the number of sentences |
| | | Lexical_diversity | Ratio of the number of distinct words to the number of total words |
| | | Group_reference | Ratio of the number of words that are connected to the group (e.g., we, us, our) to the total number of words |
| | | Individual_reference | Ratio of the number of words that are connected to individuals, namely, the first-person speaker (e.g., me, myself, I), the group (e.g., we, us, our), and the reader (e.g., you, your) to the total number of words |
| | | Self_reference | Ratio of the total number of words that are connected to the first-person speaker (e.g., me, myself, I) to the total number of words |
| | | Sentence_quantity | Total number of sentences |
| | | Verb_quantity | Total number of verbs |
| | | Word_quantity | Total number of words |
| | | Perceptual_information _and_sensory_ratio | Ratio of the number of words that are connected with perception and the senses of the total number of words |

To further improve predictive performance, validated features related to centrality and cohesion were incorporated into BM1 and BM2. These additional features were derived from the networks constructed based on the seeker's relationships during the three months preceding the project launch. The resulting enhanced models, referred to as SN-BM1 and SN-BM2, include the four major social network metrics. A random forest classifier was then employed to determine the fraudulent intent of project seekers in both the baseline and the enhanced models.

After conducting 10-fold cross-validation, the performance of all four models was evaluated and is presented in Table 10. The evaluation metrics included Accuracy, Precision, Recall, F1 Score, and AUC. Overall, the introduction of social network features led to improvements across all metrics, indicating enhanced predictive capabilities.

**Table 10. The model performance evaluation results.**

|  | Accuracy | Precision | Recall | F1 Score | AUC |
|---|---|---|---|---|---|
| **BM1 (1st Baseline Model)** | 0.6702 | 0.7028 | 0.6672 | 0.6614 | 0.7682 |
| **SN-BM1 (1st Social Network-augmented BM)** | 0.6735 | 0.7966 | 0.6997 | 0.6817 | 0.7924 |
| **BM2 (2nd Baseline Model)** | 0.7721 | 0.7808 | 0.7545 | 0.7734 | 0.8056 |
| **SN-BM2 (2nd Social Network-augmented BM)** | 0.7839 | 0.7887 | 0.7723 | 0.7816 | 0.8473 |

Furthermore, to assess the effectiveness of the newly added centrality-related features, a McNemar test was conducted. The performance of the two paired models was compared, yielding p-values of 0.0352 and 0.0221. These results indicate that the inclusion of centrality-related features significantly improved model performance. Based on the comprehensive evaluation, it can be concluded that the four major features associated with social network theory significantly contribute to fraud detection in crowdsourcing.

## Discussion

This study employs social network analysis (SNA) to investigate the behavioral signatures of fraudulent seekers in crowdsourcing contests. By systematically comparing key network metrics—degree, betweenness, and closeness centrality, clustering coefficient, and structural equivalence—between fraudulent and legitimate users, our empirical findings challenge the applicability of traditional crime network theories to digital innovation platforms and reveal a distinct fraud pattern inherent to the crowdsourcing context.

Contrary to some initial hypotheses derived from classical theories (H1b, H1c), fraudulent seekers did not occupy strategically central network positions. Instead, they exhibited significantly lower betweenness and closeness centrality than legitimate users, while confirming low degree centrality (H1a). This indicates their behavioral pattern does not rely on brokering information flow or leveraging network proximity. This finding can be explained by the task-based nature and fraud objectives within the crowdsourcing ecosystem. Traditional financial or reputational fraud often requires long-term embedding and control of social networks to build trust or manipulate information [73]. In contrast, the core objective of crowdsourcing contest fraud is to obtain high-quality solutions with minimal risk within a single project cycle. Consequently, a more effective and lower-risk behavioral pattern emerges: bypassing extensive network interactions to establish limited, point-to-point connections with high-performing solvers. This pattern naturally manifests at the macro-network level as low betweenness (not serving as a bridge) and low closeness (remaining distant from most network members). This structural position functionally reduces the node's visibility and exposure to community scrutiny, aligning more closely with the logic of short-term, goal-oriented fraud.

One of the most distinguishing findings is that all identified fraudulent seekers had a clustering coefficient of zero (H2). This means the neighbors of a fraudster are not connected to each other, forming a star-like, locally sparse connection pattern centered on the fraudster. This characteristic starkly contrasts the high cohesion (high clustering) often observed in fake review or collusive crime networks. The root cause lies in the fundamentally different relational foundations and objectives of these activities. Social e-commerce or criminal networks rely on strong trust and tight collaboration, requiring interconnections among members to solidify alliances [69,74]. In task-oriented crowdsourcing platforms, the fraudster's goal is the unilateral acquisition of intellectual property from multiple, independent solvers. If their contacted solvers were to connect and communicate among themselves, the risk of exposure—through patterns like duplicate task posting, solution similarity, or suspicious identity—would increase dramatically. Therefore, whether by deliberate avoidance or as a behavioral byproduct, maintaining a locally sparse network structure where their contacts remain disconnected serves as an effective risk isolation mechanism. This objectively results in the distinct structural fingerprint of a zero clustering coefficient.



Structural equivalence analysis confirmed that fraudulent nodes occupy similar network positions, distinct from those of legitimate users (H3). They collectively exhibit a paradoxical profile of "high surface-level activity yet low network embeddedness": they may show high project participation and connection counts [60], but these connections are of low quality (evidenced by low centrality and zero clustering). This consistent pattern reveals how fraudulent behavior adapts to and attempts to exploit the platform's credibility mechanisms. Crowdsourcing platforms often use behavioral intensity metrics like the number of posted tasks or friends as simple proxies for user trustworthiness [13]. Fraudulent seekers can inflate these metrics by posting numerous low-value tasks or adding friends indiscriminately to create a façade of compliance. However, building high-quality social capital characterized by reciprocity, trust, and deep embeddedness requires long-term, genuine cooperative investment, which conflicts with the short-term gain motive of fraud. Thus, regardless of their superficial behavioral data, their deep structural characteristics within the social network—namely, occupying sparse, peripheral, and homogeneous positions—are difficult to fake and become a stable signal for identifying fraudulent intent.

Finally, this study demonstrates the potent efficacy of SNA metrics as fraud detection tools. Incorporating the aforementioned centrality and clustering features into a machine learning model (e.g., Random Forest) led to a significant improvement in predictive performance. This indicates that moving beyond simple user attributes or activity counts to analyze a user's relational structure and position within the network can enable earlier and more accurate identification of users attempting to mask fraudulent intent with surface-level activity. For platform managers, monitoring such structural anomalies in user networks can serve as a crucial complement to existing reputation scoring systems, helping to build a more robust and multi-dimensional risk control framework.

## Conclusion and limitations

This study applied social network analysis to investigate fraudulent behavior on crowdsourcing platforms, identifying a distinct structural signature that distinguishes fraudulent seekers from legitimate users. The core findings reveal that fraudulent seekers systematically occupy peripheral network positions (low betweenness and closeness centrality), operate in locally sparse neighborhoods (zero clustering coefficient), and exhibit high structural equivalence, meaning they occupy statistically similar positions within the network.

The theoretical contribution of this study lies in refining the general understanding of fraudsters' network behavior on digital platforms. It demonstrates that fraud strategies are not static but deeply embedded within the specific interaction logic, risk structure, and profit mechanisms of a given platform. In the task-driven, competitive, and intellectual-property-focused environment of crowdsourcing, fraudulent behavior has evolved a matching network signature of "marginalization, isolation, and wide-net casting"—a pattern that fundamentally contrasts with the centrally controlling role predicted by traditional theories of network-based crime.

From a practical standpoint, the study confirms that network-derived metrics significantly enhance the performance of predictive fraud detection models. This provides platform administrators with a feasible and complementary approach—moving beyond individual attribute monitoring—to identify high-risk seekers by recognizing the relational fingerprint described above.

This study has limitations that suggest fruitful avenues for future work. First, the analysis treated projects from the same seeker as independent observations, which may affect the precision of statistical inferences. Future research could employ hierarchical models to account for this nested data structure. Second, addressing severe class imbalance through downsampling may have impacted the representativeness of the non-fraud class and the model's generalizability to natural population distributions. Utilizing alternative techniques (e.g., SMOTE) or collecting larger, more balanced datasets would strengthen future analyses. Beyond addressing these methodological points, future research should investigate the temporal dynamics of fraudulent network signatures—examining how seekers enter, maneuver within, and exit networks over time—to enable more proactive detection. Applying this SNA framework across diverse platform types would also help determine the generalizability versus context-specificity of the identified fraud pattern.



## Author contributions

**Conceptualization:** Wenjie Zhang.

**Data curation:** Wenjie Zhang.

**Formal analysis:** Wenjie Zhang.

**Funding acquisition:** Wenjie Zhang.

**Methodology:** Wenjie Zhang.

**Supervision:** Changyu Hu.

**Validation:** Changyu Hu.

**Visualization:** Zhiyuan Nong.

**Writing – original draft:** Wenjie Zhang, Zhiyuan Nong.

**Writing – review & editing:** Zhiyuan Nong, Changyu Hu.

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
