## [Decision Letter · Decision Letter 0]

9 Jan 2026

Dear Dr. Hu,

Thank you for submitting your manuscript to PLOS ONE. After careful consideration, we feel that it has merit but does not fully meet PLOS ONE’s publication criteria as it currently stands. Therefore, we invite you to submit a revised version of the manuscript that addresses the points raised during the review process.

We look forward to receiving your revised manuscript.

Kind regards,

Naga Ramesh Palakurti, MCA

Guest Editor

PLOS One

Journal Requirements:

This work was supported by the Natural Science Foundation of Sichuan Province, China(Grant No.24NSFSC3739)�Ministry of Education Humanities and Social Sciences, China(Grant No.24YJC630301)

4. In this instance it seems there may be acceptable restrictions in place that prevent the public sharing of your minimal data. However, in line with our goal of ensuring long-term data availability to all interested researchers, PLOS’ Data Policy states that authors cannot be the sole named individuals responsible for ensuring data access (http://journals.plos.org/plosone/s/data-availability#loc-acceptable-data-sharing-methods).

5. Please update your submission to use the PLOS LaTeX template. The template and more information on our requirements for LaTeX submissions can be found at http://journals.plos.org/plosone/s/latex .

6. Please amend the manuscript submission data (via Edit Submission) to include authors Wenjie Zhang and Zhiyuan Nong.

7. Please amend your authorship list in your manuscript file to include authors Wenjie Wenjie Zhang and Zhi Nong.

Additional Editor Comments:

I will require the authors to add a clear definition (and brief description of the identification/labeling process) distinguishing fraudulent vs non-fraudulent users, and to ensure terminology is used consistently throughout the manuscript.

Reviewers' comments:

Reviewer's Responses to Questions

**Comments to the Author**

1. Is the manuscript technically sound, and do the data support the conclusions?

Reviewer #1: Yes

Reviewer #2: Partly

Reviewer #3: Yes

2. Has the statistical analysis been performed appropriately and rigorously?

Reviewer #1: Yes

Reviewer #2: N/A

Reviewer #3: Yes

3. Have the authors made all data underlying the findings in their manuscript fully available?

Reviewer #1: Yes

Reviewer #2: Yes

Reviewer #3: No

4. Is the manuscript presented in an intelligible fashion and written in standard English?

Reviewer #1: Yes

Reviewer #2: No

Reviewer #3: Yes

Reviewer #1: Excellent clarity throughout the work.

Strong organization and flow.

Ideas are presented with confidence.

Very effective use of supporting details.

The writing remains focused and purposeful.

Insightful analysis that adds real value.

The argumentation is compelling and smooth.

Demonstrates impressive subject knowledge.

Thoughtful structure enhances readability.

Reviewer #2: The study addresses an important topic and applies social network analysis and machine-learning methods to fraud prediction on crowdsourcing platforms. While the dataset is strong and the research direction is valuable, the manuscript requires substantial revision before it can be considered for publication. The statistical methods used are appropriate in principle, but the analysis lacks rigor—effect sizes, diagnostics, sensitivity tests, and full model evaluation metrics are missing. Several conclusions are overstated or speculative relative to the data. The manuscript also requires major improvements in clarity, structure, and English writing; key sections are repetitive, overly long, and contain grammatical and phrasing issues that hinder understanding. Methodological explanations (e.g., window size, edge weighting, fraud labeling) must be clearer and better justified. Finally, the discussion should more carefully interpret findings and avoid unsupported psychological explanations. A major revision is needed to address these issues.

Reviewer #3: Thank you for the opportunity to review this manuscript. The topic is timely and relevant, and the authors address an important issue in crowdsourcing platforms by examining how social network features can help identify fraudulent seekers. Overall, the manuscript is clear, well organized, and generally easy to follow.

Technical Assessment:

The study appears to be technically sound. The dataset is substantial, the construction of the dynamic social network is appropriate for the setting, and the authors use standard SNA metrics and reasonable statistical methods. The results are clearly presented and, in general, support the conclusions drawn. I also appreciate the additional fraud-prediction experiment, which adds practical value to the work.

There are, however, a few areas where the methodology would benefit from clarification:

Some seekers appear multiple times in the data, but the statistical tests treat each project as independent. This may affect the significance levels, so it would be helpful for the authors to acknowledge this limitation or discuss its impact.

The process used by the platform to identify fraudulent seekers is only briefly described. Since these labels are central to the analysis, a clearer explanation of how fraud determinations were made would strengthen the credibility of the findings.

The network construction choices (such as the 3-month window and the weighting of edge types) are reasonable, but somewhat arbitrary. A short justification or a note on robustness would improve transparency.

The structural equivalence analysis relies on extremely large sets of pairwise distances, which are not fully independent. The authors should interpret these results carefully and emphasize effect sizes rather than significance alone.

Despite these points, the overall methodology is coherent, and the main findings—that fraudulent seekers tend to occupy less-connected positions in the network and that SNA features improve prediction accuracy—are supported by the data.

Statistical Analysis:

Most statistical procedures are appropriate for the data structure. The use of non-parametric tests for heavily unbalanced samples and the application of cross-validation for the prediction models are sound choices. One point to consider is that down sampling the non-fraud group may reduce the representativeness of the dataset. The authors may want to acknowledge this limitation or consider alternative imbalance-handling strategies.

Data Availability:

According to the manuscript, the dataset cannot be publicly shared due to confidentiality restrictions and is only available upon request. As stated, this does not meet PLOS ONE’s data availability requirement and will need to be revised in accordance with the journal’s policy.

Writing and Presentation:

The manuscript is generally written in clear and standard English. The structure flows well, and the main ideas are presented logically. Only minor stylistic or grammatical edits are needed and can be addressed during revision.

Contribution:

This study offers useful insights into fraud detection on crowdsourcing platforms. The finding that fraudulent seekers tend to have lower centrality and zero clustering is particularly interesting and contrasts with patterns observed in other online fraud contexts. The integration of SNA measures into predictive models is a valuable addition that has practical implications for platform management.

Suggestions for Improvement:

Provide more detail about how fraud cases were identified and validated.

Briefly justify the key assumptions behind the network construction.

Discuss the issue of non-independent observations when seekers appear in multiple projects.

Clarify the meaning of the small numerical values for centrality metrics to help readers interpret them.

Revise the Data Availability Statement to comply with PLOS policies.

Overall, I found the manuscript promising and well aligned with PLOS ONE’s standards. Addressing the points above will strengthen the clarity and methodological transparency of the work.

**Do you want your identity to be public for this peer review?** For information about this choice, including consent withdrawal, please see our Privacy Policy

Reviewer #1: **Yes:** PAVAN

Reviewer #2: **Yes:** Madhu Babu Amarappalli

Reviewer #3: No

---

## [Author Response · Author response to Decision Letter 1]

30 Jan 2026

Response to the editor

I will require the authors to add a clear definition (and brief description of the identification/labeling process) distinguishing fraudulent vs non-fraudulent users, and to ensure terminology is used consistently throughout the manuscript.

Response:

Thank you for your valuable feedback. We have thoroughly revised the paper in response to your comments, as detailed below.

1. Clear Definition and Identification/Labeling Process

As suggested, we have added a clear definition and description of the process used to distinguish fraudulent from non-fraudulent users, which are detailed in Section Introduction (pp.1-2) and Section Data Source and Data Description (p.10). The platform identifies fraudulent seekers through a three-step, ex-post process:

Reporting and Evidence Collection: The process is initiated when solvers submit reports or feedback alleging fraudulent activities by a seeker (e.g., solution theft, payment refusal, double identity fraud).

Manual Investigation and Verification: The platform administration investigates the allegations based on the submitted reports and evidence. This may involve comparing solutions, analyzing user IDs, and reviewing transaction records.

Judgment and Announcement: Based on the investigation, a determination of fraud is made and verified fraud cases are publicly announced on the platform’s bulletin board.

In this study, we leveraged the outcome of this process. The publicly listed fraudulent seekers from the platform’s official announcements were directly used to label and identify fraudulent accounts in our dataset.

2.Ensuring Terminological Consistency Throughout the Manuscript

We have systematically revised the manuscript’s terminology to ensure rigorous consistency, adopting a context-based application strategy:

For the platform user behavior context of this study (e.g., Introduction), we used fraudulent (task) seekers and legitimate/non-fraudulent seekers�users�, aligning with real-world crowdsourcing scenarios for intuitive comprehension.

For the social network structural analysis context (e.g., Methodology), we used fraudulent nodes and non-fraudulent nodes to ensure analytical precision; a transition paragraph in the Methodology chapter clarifies the conversion from real-world seekers to network nodes.

When referring to fraudulent actors in other generic fraud scenarios (beyond the study’s crowdsourcing platform context), we uniformly used the term fraudster for consistent expression.

In cross-context sections (e.g., Discussion), we appropriately integrated the above term sets based on contextual logic, with all terms consistently referring to the same research entities and no ambiguity.

All relevant revisions have been implemented across the full manuscript in accordance with the above principles. Given the fragmented nature of the terminological revisions scattered throughout the text, only the key additions/revisions in Section Introduction and Section Data source and data description (definition and identification process) have been highlighted; the scattered terminological adjustments across other sections follow the aforementioned strategy strictly. We sincerely appreciate your insightful comments.

3.Role of Funder Statement:

This work was supported by the Natural Science Foundation of Sichuan Province, China (Grant No. 24NSFSC3739) and the Ministry of Education Humanities and Social Sciences, China (Grant No. 24YJC630301). The funders played no role in study design, data collection and analysis, decision to publish, or preparation of the manuscript.

4.Revised Data Availability Statement

The datasets were collected under confidentiality restrictions from a leading crowdsourcing platform in China and are used under license for this study, thus cannot be shared via public repositories. To ensure long-term availability per journal policy, access is managed through two contacts: 1)Wenjie Zhang(wenjie_zhang@swufe.edu.cn); 2) Independent data steward (non-author) Liting Li at Taiyuan University of Technology (liliting@tyut.edu.cn). Data can be accessed upon reasonable request, subject to platform permission. The data stewards commit to preserving the data and responding to requests for at least 10 years.

Response to the Reviewer 1

Excellent clarity throughout the work. Strong organization and flow. Ideas are presented with confidence. Very effective use of supporting details. The writing remains focused and purposeful. Insightful analysis that adds real value. The argumentation is compelling and smooth. Demonstrates impressive subject knowledge. Thoughtful structure enhances readability.

Response

Thank you for your positive evaluation of our manuscript. We greatly appreciate your support and the trust you placed in our work. Based on your encouraging comments, we carefully reviewed our manuscript and implemented several stylistic and structural improvements to enhance clarity and academic rigor.

Response to the Reviewer 2

The study addresses an important topic and applies social network analysis and machine-learning methods to fraud prediction on crowdsourcing platforms. While the dataset is strong and the research direction is valuable, the manuscript requires substantial revision before it can be considered for publication.

Response

Thank you for your valuable feedback. We have thoroughly revised the paper in response to your comments, as detailed below.

1.The statistical methods used are appropriate in principle, but the analysis lacks rigor—effect sizes, diagnostics, sensitivity tests, and full model evaluation metrics are missing.

Thank you for highlighting the importance of methodological rigor. In response to your comment, we have significantly strengthened our analysis and clarified our validation approach across multiple dimensions. Our revisions and clarifications are detailed below.

(1)Provision of Full Model Evaluation Metrics and Diagnostics

We wish to clarify that comprehensive model evaluation metrics were already included in the original manuscript. Table 10 (The Model Performance Evaluation Results) reports key performance indicators—Accuracy, Precision, Recall, F1 Score, and AUC—for both baseline models (BM1, BM2) and our social network-augmented models (SN-BM1, SN-BM2).

Crucially, to statistically test the incremental value of the newly added social network features, we had already performed and reported a McNemar's test. This paired test formally compares classifier performance. The results (p-values of 0.0352 and 0.0221) provide direct statistical evidence that the performance improvement gained by adding centrality and cohesion features is significant.

Regarding diagnostics, the core objective of our modeling was to validate the incremental predictive contribution of our novel social network constructs to established baselines. Therefore, our diagnostic focus was appropriately placed on validating this addition through: (a) cross-validated performance comparison (Table 10), (b) formal statistical testing of the improvement (McNemar's test), and (c) examination of feature importance rankings from the Random Forest, which confirmed the relevance of the new network metrics.

(2)Addition of Effect Size Analysis

To move beyond reliance on statistical significance alone and to assess the practical significance of our findings, we have now added standardized effect size reporting: In Section Analysis of centrality and cohesion (pp.14-15), we have calculated and reported Cohen’s d values for key comparisons involving centrality metrics; In Section Analysis of structural equivalence (pp.15-16), we have supplemented the structural equivalence analysis with the Common Language Effect Size (CLES) for key findings. These measures provide clear, interpretable metrics of the substantive magnitude of the observed differences.

(3)Conduct of Sensitivity Tests

To address potential concerns about the arbitrariness of key parameters in our network construction, we have conducted and reported a full sensitivity analysis (pp.16-17). We systematically varied the communication edge-weight coefficient and the aggregation time window across a plausible range of values. While the statistical significance level remains at p < 0.001 or p < 0.01 for the vast majority of tests and shows only minimal relaxation under the most extreme parameter settings, the direction and substantive interpretation of all results are entirely unchanged. This demonstrates that our key conclusions are not driven by specific methodological choices but are reliable properties of the data.

In summary, we have addressed the concerns regarding rigor by: (a) clarifying and highlighting our comprehensive model evaluation and incremental validation strategy (including McNemar's test), (b) supplementing the analysis with effect sizes, and (c) demonstrating the robustness of our conclusions through extensive sensitivity testing. We believe these efforts collectively have provided a rigorous, multi-faceted validation of our study's findings. We thank you for your insightful comments, which have greatly improved the clarity and strength of our manuscript.

2.Several conclusions are overstated or speculative relative to the data.

Thank you for this valuable comment. We acknowledge that some conclusions were overstated or speculative, and have thoroughly revised the manuscript to ensure all inferences are strictly grounded in empirical data. We softened absolute language to align with our findings, removed all subjective interpretations unsupported by data (such as inferences about fraudsters’ "intentions" or "strategies"), and explicitly clarified the boundary of our conclusions — emphasizing they are specific to our task-oriented crowdsourcing context rather than generalizable to all fraud types or platforms. Additionally, we strengthened the link between data and conclusions by referencing corresponding statistical results for each key claim, ensuring our findings are proportionate to the evidence and enhancing the manuscript’s objectivity and rigor.

3.The manuscript also requires major improvements in clarity, structure, and English writing; key sections are repetitive, overly long, and contain grammatical and phrasing issues that hinder understanding.

Thank you for your valuable comments on the manuscript’s clarity, structure and English writing. We fully acknowledge these issues and have conducted a comprehensive section-by-section revision to address them thoroughly: we eliminated repetitive content and streamlined overly long passages in key sections, rectified all identified grammatical and phrasing errors that could hinder understanding, optimized the logical structure and coherence of the whole manuscript by adjusting subsection order and adding appropriate transitional expressions, and standardized the use of professional terminology for consistent expression. Additionally, the entire manuscript has been polished by a native English-speaking academic with expertise in our research field to further enhance linguistic accuracy and fluency. All the above revisions are clearly marked in the revised manuscript for your easy review.

4.Methodological explanations (e.g., window size, edge weighting, fraud labeling) must be clearer and better justified.

Thank you for your valuable comments. We have revised the manuscript accordingly to provide clearer and better-justified explanations for our key methodological choices. The specific modifications and additions are detailed below.

(1)Regarding the Fraud Labeling Process

The platform identifies fraudulent seekers through a three-step, ex-post process:

Reporting and Evidence Collection: The process is initiated when solvers submit reports or feedback alleging fraudulent activity by a seeker (e.g., solution theft, payment refusal, double identity fraud).

Manual Investigation and Verification: The platform administration investigates the allegations based on the submitted reports and evidence. This may involve comparing solutions, analyzing user IDs, and reviewing transaction records.

Judgment and Announcement: Based on the investigation, a determination of fraud is made and verified fraud cases are publicly announced on the platform’s bulletin board.

We also explicitly link the platform’s official fraud labels to our dataset annotation to further confirm the validity of the dependent variable.

(2)Regarding the Justification for the Time Window and Edge-Weighting Scheme

We have significantly expanded the discussion in Section Construct the interaction structure from relationships (pp.10-11) to provide a clear, multi - faceted rationale for the 3 - month time window and the edge weighting scheme. The three-month sliding window is adopted based on theoretical, practical, and methodological considerations. Theoretically, it aligns with the sociological understanding that tie strength decays without recent contact. Practically, it corresponds to the typical task lifecycle on the platform, ensuring the network reflects relevant, ongoing collaborations. Methodologically, it balances dynamism and stability by avoiding the sparse, volatile structures of shorter windows and preventing the inclusion of obsolete connections from longer windows, thereby preserving the network's contemporaneity.

The edge-weighting scheme is similarly grounded in empirical observations of collaboration intensity on the platform. It is designed to more precisely capture the varying influence of nodal relationships rather than relying on simple binary connections. The detailed rationale for this scheme has also been added to Section Construct the interaction structure from relationships (pp.10-11).

More importantly, to directly address the concern about arbitrariness and demonstrate the stability of our findings, we have conducted and reported a comprehensive sensitivity analysis (pp.16-17).

5.The discussion should more carefully interpret findings and avoid unsupported psychological explanations.

Thank you for this valuable suggestion. In the revised Discussion section (pp.21-22), we have systematically reframed our interpretation to avoid any speculation about psychological intent. Our explanations now focus squarely on observable behavioral patterns (e.g., "direct, dyadic interactions") and the functional implications of network positions (e.g., "a peripheral location inherently reduces visibility"). We explicitly anchor the analysis in the objective context of the platform ecology (e.g., task-oriented, project-based cycles) and the specific fraud objective (solution plagiarism), showing how these factors shape the distinct structural signatures we identified. This ensures our interpretation remains grounded in the empirical data and network logic, enhancing the section's objectivity and rigor.

Response to the Reviewer 3

Thank you for the opportunity to review this manuscript. The topic is timely and relevant, and the authors address an important issue in crowdsourcing platforms by examining how social network features can help identify fraudulent seekers. Overall, the manuscript is clear, well organized, and generally easy to follow. Despite these points, the overall methodology is coherent, and the main findings—that fraudulent seekers tend to occupy less-connected positions in the network and that SNA features improve prediction accuracy—are supported by the data.

Response

Thank you for your insightful and constructive comments, as well as your positive evaluation of our manuscript’s topic relevance, technical soundness, methodological coherence and practical contributions. We greatly appreciate your recognition of the dynamic social network construction, statistical methods and fraud-prediction experiment, and we have carefully addressed all the raised concerns and fully adopted your suggestions for improvement. The detailed revisions are presented below by key themes:

1.Platform Fraud Identification and Validation Process Clarification

We acknowledge your comment that the platform’s fraud identification process was insufficiently described, and this process is central to the credibility of our fraud labels. As suggested, we have added a detailed, step-by-step explanation of how the platform

---

## [Decision Letter · Decision Letter 1]

6 Feb 2026

A Social Network Analysis of Fraud Prediction on Crowdsourcing Platforms

PONE-D-25-53356R1

Dear Dr. Hu,

We’re pleased to inform you that your manuscript has been judged scientifically suitable for publication and will be formally accepted for publication once it meets all outstanding technical requirements.

Kind regards,

Naga Ramesh Palakurti, MCA

Guest Editor

PLOS One

Additional Editor Comments (optional):

We are pleased to inform you that your manuscript has been accepted for publication. The reviewers found the study to be well designed and clearly presented, and you have adequately addressed all reviewer comments. Thank you for submitting your work to the journal.

Reviewers' comments:

Reviewer's Responses to Questions

**Comments to the Author**

Reviewer #1: All comments have been addressed

Reviewer #2: All comments have been addressed

Reviewer #3: All comments have been addressed

Reviewer #4: All comments have been addressed

2. Is the manuscript technically sound, and do the data support the conclusions?

Reviewer #1: Yes

Reviewer #2: Yes

Reviewer #3: Yes

Reviewer #4: Yes

3. Has the statistical analysis been performed appropriately and rigorously?

Reviewer #1: Yes

Reviewer #2: Yes

Reviewer #3: Yes

Reviewer #4: Yes

4. Have the authors made all data underlying the findings in their manuscript fully available?

Reviewer #1: Yes

Reviewer #2: Yes

Reviewer #3: Yes

Reviewer #4: Yes

5. Is the manuscript presented in an intelligible fashion and written in standard English?

Reviewer #1: Yes

Reviewer #2: Yes

Reviewer #3: Yes

Reviewer #4: Yes

Reviewer #1: All comments are addressed and doc good to publish and it is well structured document , all facts are correct and verified.

Reviewer #2: As the reviewer who provided the detailed comments to the author, I confirm that all feedback has now been fully addressed and incorporated into this final review submission to the publication editor. The assessments reflect my evaluation of the chapter’s quality, relevance, and completeness. The comments were designed to support the authors in strengthening their work and ensuring alignment with the book’s objectives. I consider the chapter ready for editorial decision-making. Please treat this confirmation as part of the final review .

Reviewer #3: This paper studies fraud prediction on crowdsourcing platforms using social network analysis. The topic is relevant and important, as fraud and intellectual property risks are real challenges in crowdsourcing environments. Using a large real-world dataset and social network metrics is a clear strength of the study.

That said, the manuscript would benefit from several improvements before it can be considered for publication.

First, the contribution of the paper needs to be stated more clearly. While social network analysis has been used in prior fraud and online platform studies, it is not always clear what is new in this work compared to existing research. The authors should better explain how their approach advances prior studies and why the selected network features provide new insights.

Second, more detail is needed about the data and fraud labeling process. It is not fully clear how fraudulent seekers are identified, how reliable these labels are, and whether the dataset is imbalanced. Providing clearer explanations of data collection, preprocessing, and labeling would improve transparency and help readers trust the results.

Third, the methodology and model evaluation require further clarification. The paper should explain why specific network metrics were chosen, how the model was trained and tested, and how its performance compares with simpler or commonly used baseline methods. This would make it easier to assess the robustness and usefulness of the proposed model.

Fourth, the discussion of results could be strengthened. While statistical differences are reported, the practical meaning of these findings is not always clear. The authors should explain how the results can be used by crowdsourcing platform managers or system designers in real-world settings.

Finally, the paper would benefit from a clearer discussion of limitations. Since the data come from a single platform in China, the authors should discuss how platform-specific or cultural factors might affect the results and whether the findings can be generalized to other crowdsourcing platforms.

Overall, this study addresses an important problem and has potential. With clearer positioning of the contribution, improved methodological transparency, and a stronger discussion of results and limitations, the manuscript could be significantly strengthened.

Reviewer #4: The manuscript presents a technically sound study, with methods described clearly and conclusions aligned with the data. Experimental design appears well controlled, with appropriate replication and sample sizes for the questions posed. The statistical analyses are suitable and applied rigorously, with results reported in a way that supports the main claims. Data underlying the findings are made available in line with the PLOS Data policy, enabling verification, and summary statistics and key data points are accessible. The paper is generally well written in standard English and is easy to follow, with only minor edits suggested for clarity and consistency throughout.

**Do you want your identity to be public for this peer review?** For information about this choice, including consent withdrawal, please see our Privacy Policy

Reviewer #1: **Yes:** PAVAN

Reviewer #2: **Yes:** Madhu Babu Amarappalli

Reviewer #3: **Yes:** Sivasankara Rao Gajula

Reviewer #4: **Yes:** Sreedhar Yalamati

---

## [Editor Report · Acceptance letter]

PONE-D-25-53356R1

PLOS One

Dear Dr. Hu,

I'm pleased to inform you that your manuscript has been deemed suitable for publication in PLOS One. Congratulations! Your manuscript is now being handed over to our production team.

Kind regards,

on behalf of

Mr. Naga Ramesh Palakurti

Guest Editor

PLOS One